# Variation among the Complete Chloroplast Genomes of the Sumac Species *Rhus chinensis*: Reannotation and Comparative Analysis

**DOI:** 10.3390/genes13111936

**Published:** 2022-10-25

**Authors:** Yujie Xu, Jun Wen, Xu Su, Zhumei Ren

**Affiliations:** 1School of Geosciences, Qinghai Normal University, Xining 810008, China; 2School of Life Science, Shanxi University, 92 Wucheng Rd., Taiyuan 030006, China; 3Department of Botany, National Museum of Natural History, Smithsonian Institution, Washington, DC 20013, USA; 4Academy of Plateau Science and Sustainability, Qinghai Normal University, Xining 810016, China; 5Key Laboratory of Biodiversity Formation Mechanism and Comprehensive Utilization of Qinghai-Tibetan Plateau in Qinghai Province, School of Life Sciences, Qinghai Normal University, Xining 810008, China

**Keywords:** *Rhus chinensis*, chloroplast genome, genomic structure, comparative analysis, phylogeny

## Abstract

The sumac *Rhus chinensis* Mill. is an economically and ecologically important shrub or tree species in the family of Anacardiaceae with a wide distribution in East to Southeast Asia. We assembled the complete chloroplast genome of 159,187 bp in length and the GC content of 37.8%. The genome encoded 132 genes, including 86 protein-coding genes, 37 tRNA genes, 8 rRNA genes, and 1 pseudogene, and 77 SSRs were identified as well as the interval regions, totaling 46,425 bp in length. The mauve alignment revealed one gene rearrangement among the *Rhus* species. All the SSRs were divided into five types, most of which consisted of mono- and tri- repeat motifs. Our genome exhibited the longest size and more annotated genes compared to the three other genomes of *R. chinensis* reported in GenBank. We also discovered some relatively highly variable regions in the complete chloroplast genomes of the *Rhus* species. The ML phylogenetic analysis of the available chloroplast sequences of the Anacardiaceae well supported the monophyly of each tribe and each genus; the tribe Rhoideae was close to the tribe Anacardiaceae with a high support of 100%, and they then grouped with the tribe Spondiadeae. *R. chinensis* was sister to *R. potaninii*, and they then grouped with the species *R. typhina*.

## 1. Introduction

The sumac genus *Rhus* (Anacardiaceae) contains c. 35 species mainly distributed in temperate and sub-tropical regions [1] and is widely recognized in the north temperate zone for its adaptability to various climate and soil conditions [2,3]. *R. chinensis* is a common deciduous sumac tree that is widely distributed in Asia, including China, India, Vietnam, Korea, and Japan [4]. This species is an important medicinal plant containing various pharmacologically active constituents, which have been used for medicinal purposes such as anticancer, antiviral, antimicrobial, and anti-inflammatory as well as being used as a revegetation plant for ecological restoration due to its cold tolerance and easy multiplication [5,6,7,8]. In particular, as the primary host plant of several species of Chinese sumac aphids, *R. chinensis* hosted the aphids which induced galls, which are a source of traditional Chinese medicine [9]. The galls are economically very important because they are rich in tannins, which were extracted and synthesized to be utilized earlier in the rubber industry and for improving leather quality, while at present they are used in various fields, e.g., medicine, food, dye, chemical, and military industries [9,10]. However, in recent decades, their saplings have been repeatedly destroyed by anthropogenic activities, which severely restricted their natural regeneration. The number of wild *R. chinensis* plants is decreasing, and their natural distribution range is gradually shrinking [11].

Most of the previous studies on *R. chinensis* mainly focused on its pharmacology, phytochemical, biological activities, biological features, and biogeographic diversification [12,13,14,15,16]. In the case of the molecular analysis, the chloroplast fragments (*ndhF*, *rbcL*, *trnL-F*, and *trnC-trnD*) and nuclear ITS sequences were applied as markers to study the genetic diversity and phylogenetic relationships of the *Rhus* species [17,18,19,20]. The analysis of the nuclear ITS region and the chloroplast fragments indicated that *Rhus* was monophyletic, but the relationships among these genera of the *Rhus* complex were not well resolved [18,19,20]. Thus, obtaining more information on the genome of the *Rhus* species is essential to further study their delimitation, genetic diversity, phylogenetic relationships, and evolutionary history [21].

The chloroplasts are well known to be key organelles in plants with crucial functions in photosynthesis and biosynthesis [22]. Moreover, compared with nuclear and mitochondrial genomes, the chloroplast genomes in angiosperms are highly conserved in structure, gene content, and organization [23,24]. In particular, for its exhibiting variations at the intra-species level, the genome has been widely used for the study of systematics and evolution among different populations [25,26]. In recent years, more and more complete chloroplast genomes in plants have been reported [27,28,29], which provide more valuable information and screen more new markers with potentially high resolution for more comprehensive research in the future to better resolve the phylogenetic relationships and intraspecific diversity of the *Rhus* species. At present, there were three complete chloroplast genomes of *R. chinensis* reported [21,30,31]. However, we found that these three genomes showed large differences in both sequence length, from 149,011 bp to 158,809 bp, and number of genes, from 126 to 130, which is very unusual for different accessions of the same species. Hence, it is important to obtain the high-quality genome sequences, especially in the case of organellar genomes, for the analysis of genetic diversity and phylogenetics [29].

In the present study, we sequenced and assembled a complete chloroplast genome of *R. chinensis* and analyzed its nucleotide composition, gene organization, and structure, especially in comparison with the three reported chloroplast genomes of *R. chinensis* and other *Rhus* species downloaded from GenBank (https://www.ncbi.nlm.nih.gov/GenBank, accessed on 29 September 2021). We aimed to accurately assemble and validate the complete chloroplast genome of the species *R. chinensis* in order to further examine the genetic variation among these chloroplast genomes of *R. chinensis* accessions and other *Rhus* species. Furthermore, the phylogeny of the Anacardiaceae species was constructed for the purpose of clarifying the relationships between different species, genera, and tribes.

## 2. Materials and Methods

### 2.1. Sample, DNA Extraction, and Sequencing

We collected fresh leaves of *R. chinensis* at Wufeng county (30°11′10.514″ N, 111°5′42.342″ E; alt. 824 m), Hubei, China, on 6 October 2014. The leaf samples were dried in a timely manner using silica gel, and the total genomic DNA of *R. chinensis* was extracted using the Plant Genomic DNA Kit (TIANGEN) following its instructions. The quality and quantity of the extracted DNA were examined using Nanodrop 2000. The qualified total DNA was used for library construction and sequencing by using the shotgun genome skimming method on an Illumina HiSeq 4000 platform [32]. The paired-end (PE) reads of 2 × 150 bp (insert size of 400 bp) were generated and the qualified DNA sequences were obtained after filtering out low-quality and adapter-contaminated reads. A total of 8.4 G clean reads was used for assembling the genome. The specimen was stored at the Herbarium of School of Life Science, Shanxi University, China (voucher no: Ren_P1966).

### 2.2. Chloroplast Genome Assembly and Annotation

The clean reads were decompressed in Terminal, and then the complete chloroplast genome was assembled using the de novo method by the program GetOrganelle with kmers 21, 45, 65, 85, and 105 [33]. Additionally, we assembled the genome by mapping it with the chloroplast genome of *R. chinensis* (KX447140) and *Pistacia weinman**niifolia* (NC_037471) as references. The annotations of chloroplast genome were conducted by the PGA software [34] and the Geneious Prime software (version 11.0.3; https://www.geneious.com, accessed on 29 September 2021) using the reported genome sequences of *Rhus* species as references (GenBank accession Nos: KX447140, MF351625, and NC_037471). In addition, we searched the homologous sequences of the unannotated or ambiguous region by BLAST in GenBank to attempt annotate these regions. Start/stop codons and intron/exon borders of protein coding genes with mistakes were detected by translating the sequences. In addition, the transfer RNA (tRNA) genes were verified by the online tRNAscan-SE v2.0 software [35]. The physical map of the complete chloroplast genome of *R. chinensis* was generated utilizing the OGDRAW program (https://chlorobox.mpimp-golm.mpg.de/OGDraw.html, accessed on 1 December 2021) [36] to demonstrate its structural characteristics.

### 2.3. Sequence Analysis

We performed the statistics on the general information of the complete chloroplast genome of *R. chinensis*, including the total genome length, the gene number and size, the base content, and the lengths of exons and introns in genes using the Geneious Prime software (version 11.0.3; Created by the Biomatters development team Copyright; New Zealand).

The relative synonymous codon usage (RSCU) ratio was obtained by the programs MEGA7.0 and CodonW. MISA was used to detect the microsatellites, also called simple sequence repeats (SSRs), i.e., mono-, di-, tri-, tetra-, penta-, and hexa-nucleotide repeats, with thresholds of ten, six, and three repeat units for mononucleotide SSRs, di- and tri-nucleotide SSRs, and tetra-, penta-, and hexa-nucleotide SSRs, respectively [37].

### 2.4. Genome Comparison

The complete chloroplast genome of *R. chinensis* was compared with those of its related species *Rhus* to detect the genomic variation. Multiple genome alignment among complete chloroplast genomes of *Rhus* species was conducted through the Mauve program to detect evolutionary events such as rearrangement and inversion.

The borders of large single-copy (LSC), small single-copy (SSC), and inverted repeat (IR) regions were visually displayed and compared among *Rhus* species using BLAST, Find Repeats regions, and IRscope (http://irscope.shinapps.io/irapp/, accessed on 10 December 2021).

We used two datasets, the sequences of four complete chloroplast genomes of *R. chinensis*, and the other sequences of six complete chloroplast genomes of *Rhus* species excluding *R. chinensis* (accession Nos: KX447140 and MF351625) to detect the hotspots of intraspecific and species divergence, and analyze sequence divergence using the MAFFT online service (https://mafft.cbrc.jp/alignment/server/, accessed on 2 January 2022). We also analyzed the nucleotide diversity (Pi) using the DnaSP software with the step size of 200 bp and the window length of 800 bp [38].

### 2.5. Phylogenetic Reconstruction

All the protein-coding genes from the complete chloroplast genomes were used to construct the phylogenetic tree of the family Anacardiaceae. The detailed species information used in this study is shown in Table 1. Each protein-coding gene was aligned using MAFFT version 7 [39,40] and then concatenated as a dataset by Geneious v11.0.3 for phylogenetic analysis.

The phylogenetic relationship of 21 Anacardiaceae species including six genera was constructed with two Burseraceae species, namely *Commiphora gileadensis* and *Boswellia sacra*, as outgroups. We performed the construction of the maximum likelihood tree using the RAxML program under the GTRGAMMA model with 1000 bootstrap replicates [41].

## 3. Results and Discussion

### 3.1. General Features of the Complete Chloroplast Genome of R. chinensis

The assembled complete chloroplast genome of *R. chinensis* totaled 159,187 bp in length, comprising an LSC region of 87,961 bp, an SSC region of 18,522 bp, and a pair of IR (IRA/IRB) regions of 26,506 bp. The genome contained 132 genes with 86 protein-coding genes, 37 tRNAs, 8 rRNAs, and 1 pseudogene, among which 60 protein-coding genes were in the LSC region, and 11 in the SSC region. All the protein-coding genes started with the common initiation codon ATG. There were 22 tRNAs and 1 tRNA in the LSC and SSC region, respectively. The IR region duplicated eight protein-coding genes and seven tRNAs. All rRNA genes were located in the IR region. The location and organization of the complete chloroplast genome of *R. chinensis* are shown in Table 2, and the diagram is shown in Figure 1. The genome sequence was submitted to GenBank with the accession No. OP326720.

The GC content of the complete chloroplast genome totaled 37.8%, which was much lower than the AT content (about 62%). The GC content (42.9%) in the IR regions was the highest among those of the overall genome (37.8%), the LSC (35.9%), and the SSC regions (32.6%), which may result from the presence of GC-rich rRNAs and tRNAs in these regions. The statistics of the base composition showed that the third codon position had the highest AT content (69.5%), whereas the first codon position had the lowest AT content (53.6%), with 62.2% at the second codon position. Our results are consistent with the previous reports on the plant chloroplast genomes [31,42].

There were 21 one-intron-containing genes, including 13 protein-coding genes and 8 tRNA genes, whereas 2 protein-coding genes, i.e., *ycf3* and *clpP*, possessed two introns (Table 3). Among the 21 genes, the shortest intron was found in the *trnL-UAA* gene with 450 bp in length, whereas the longest was found in the *trnK-UUU* gene with 2599 bp and contained the protein-coding gene *matK*, which are the general characteristics in other land plants [43]. The gene *rps12* was trans-spliced with the duplicated 3′ end in the IRs and the 5′ end located in the LSC region, as previously reported in other plants [44].

The results of the relative synonymous codon usage (RSCU), one of the commonly used parameters to measure codon usage bias, reveal that the 86 protein-coding genes of *R. chinensis* plastome included 26,654 codons (Figure 2). Among these codons, Leu was the most abundant amino acid, at 2805 (10.5%), while Cys was the least abundant, at 312 (1.2%). The 31 codons with RSCU values >1 mostly ended with A/T (U), while the 31 other codons having RSCU values <1 mostly ended with G/C. The remaining two, Met and Tyr, showed no biased usage (RSCU = 1).

We performed the SSR analysis using the MISA program, and the results are shown in Figure 3. A total of 77 perfect microsatellites were identified in the chloroplast genome of our *R. chinensis* sample, which included 52 mononucleotides, 4 dinucleotides, 7 trinucleotides, 12 tetranucleotides, and 2 pentanucleotides, whereas the hexanucleotide repeats were not detected (Figure 3A). Among these SSRs, all mononucleotides were composed of A/T and all dinucleotides were composed of AT/TA, whereas the rest of the SSRs also had a high A/T content. Our findings are consistent with the previous reports, which showed that the SSRs in the chloroplast genome were composed of polyA or polyT repeats, while they rarely contained tandem G or C repeats. In the total SSRs loci, the repeats located in the LSC region were much higher (68.83%) than those in the SSC (18.18%) and IR (12.99%) regions (see Figure 3B). Most of these SSRs (75.4%) were located in intergenic regions (IGS), and the rest (24.6%) were located in protein-coding regions and tRNA genes (eight SSRs in introns).

### 3.2. Comparison among R. chinensis Accessions

We downloaded the three complete chloroplast genomes of *R. chinensis* from GenBank, which were from Gangwon (accession No. KX447140) in Korea, Shandong (accession No. MF351625) and Anhui (accession No. MG267385) in China, and examined their variation in the different accessions of the *R. chinensis* species. By comparison, we found great differences among these genomes, not only in the nucleotide variation but also in the length and gene numbers. Therefore, we re-annotated all the complete chloroplast genomes downloaded from GenBank to avoid the mistakes from the annotation. As a result, the sample with GenBank (accession No. MG267385) obtained more annotated genes. The diversity and variation of four complete chloroplast genomes of *R. chinensis* are shown in Table 4. We can see that the genomes ranged from 149,011 bp to 159,187 bp in length with the longest difference being of 10,176 bp between our current sample and the accession with No. KX447140. Our current sample exhibited the longest genome size and the most annotated genes compared to the other three accessions, and the sample with accession No. KX447140 had the shortest genome size and the least genes. All the chloroplast genomes displayed the quadripartite structure with a pair of IRs (16,602–26,550 bp) separated by the LSC (87,045–97,246 bp) and SSC (18,522–18,674 bp) regions, while they exhibited obvious differences in genome size and in the lengths of the LSC and IR regions. All four individuals varied in the number of genes, with the total gene number ranging from 126 to 132, the total protein-coding genes ranging from 82 to 86, and the total tRNA genes ranging from 36 to 37. All four complete chloroplast genomes contained eight rRNA genes. The overall GC content in each complete chloroplast genome was approximately 37.9%.

We examined the gene rearrangement of four complete chloroplast genomes of *R. chinensis* by alignment using the Mauve software implemented in Geneious (Figure 4A). All the four genomes had the same gene arrangement except for the missing regions, and the differences in genome length and in number of genes were mainly from the IR region. The gene organization and order in the IR region are shown in Figure 4B. Only one *rpl2* intron-containing gene was observed in the IR region of the accessions KX447140 and MF351625, while the four genes *rpl2*, *rpl23*, *trnI-CAU*, and *ycf2* had two copies in the two *R. chinensis* samples, our current study, and the accession No. MG267385. Furthermore, two more genes, namely *rpl22* and *rps19*, were only annotated in our sample. The lengths of exons and introns in intron-carrying genes also exhibited some slight changes among four complete chloroplast genomes, and the length variations in the introns were greater than those in the exons (Table 3). In general, the exon lengths were conserved in the four *R. chinensis* individuals except for the *ndhA* and *atpF* genes.

We compared the nucleotide diversity in the total, LSC, SSC, and IR regions of the four complete chloroplast genomes of *R. chinensis* (Table 4). In total, 1436 variable sites (0.89%), including 93 parsimony-informative sites (0.06%), were examined in the chloroplast genomes. The LSC and SSC regions contributed 29 and 20 informative sites, respectively, while the IR regions only contributed 11 informative sites. Among these regions, the IR regions exhibited the least nucleotide diversity (0.00156) and the SSC region exhibited the highest divergence (0.00783).

Genome-wide comparative analyses among the four individuals of *R. chinensis* showed that the non-coding region exhibited a higher nucleotide divergence than the coding region and the SC region with higher divergence than the IR regions, respectively. The peak with the highest degree (Pi: 0.035) of difference was located between the *rps3* and *trnL* genes. By sequence alignment, there were missing fragments between the two *rpl22* and *ycf2* genes, including four genes: *rps19*, *rpl2*, *rpl23*, and *trnI-CAU* (Figure 5A). Furthermore, the nucleotide diversity (pi) value within 800 bp was calculated to estimate the sequence divergence level. The pi values varied from 0 to 0.035 for the four *R. chinensis* chloroplast genomes. With the exception of the *rps3-trnL* region, we identified 10 hotspot regions for genome divergence, 7 of which (*trnK-rps16*, *trnS-trnG*, *trnE-trnT*, *trnG-psaB*, *trnF-ndhJ*, *accD-cemA*, and *psbL-petL*) were located in the LSC region, and 3 (*ndhF-rpl32-trnL*, *ccsA-ndhD*, and *ycf1*) in the SSC region, while only 1 *ycf1* was located in the coding region. Interestingly, the highly variable regions of *R. chinensis* were different from those previously reported for designing phylogenetic trees and the species identification of *Rhus*, such as *trnL-trnF*, *trnC-trnD*, and *rbcL* [17,18,19,20]. Thus, by analyzing the variable regions of the chloroplast genomes, we were able to identify some molecular markers, and these hotspot regions could be utilized as potential markers to reconstruct the phylogeny and plant identification.

### 3.3. Comparison among the Rhus Species

We downloaded the four complete chloroplast genomes of *Rhus potaninii* and *R. typhina* from GenBank, which were from Beijing (accession No. MN866893), Shaanxi (accession No. MT230556), Shandong (accession No. MT083895), and an unknown location (accession No. MN866894). Combined with *R. chinensis*, we conducted a comparative analysis to test their features and variation excluding two *R. chinensis* sequences (accession No. MF351625 and KX447140) due to their large missing regions (the sequences need to be confirmed). All the six *Rhus* chloroplast genomes ranged from 160,254 bp to 158,809 bp in length and contained a pair of inverted repeat regions (IRs: 26,475–26,550 bp), which were separated by a small single-copy region (SSC: 18,522–19,453 bp) and a large single-copy region (LSC: 87,045–87,789 bp). The number of the total genes in the chloroplast genomes ranged from 130 to 133, the number of total protein-coding genes ranged from 85 to 86, and all the genomes contained 37 tRNA genes and 8 rRNA genes, among which the 2 protein coding genes, namely *rpl22* and *rps19*, were present in our *R. chinensis*, *R. potaninii* (accession Nos: MN866893 and MT230556), and *R. typhina* (accession No. MN866894 and MT083895) samples, but not in *R. chinensis* with accession No. MG267385.

In order to determine the level of sequence divergence, we compared nucleotide diversity in the whole genome, the LSC, SSC, and IR regions of six complete chloroplast genomes of the *Rhus* species. As shown in Table 4, in total, we examined 2931 variable sites (1.8%), including 1621 parsimony-informative sites (0.99%). The LSC and SSC regions contributed 1075 and 435 informative sites, respectively, while the IR regions only contributed 61 informative sites. The nucleotide diversity (Pi) among six *Rhus* species’ chloroplast genomes is 0.00841. The analysis of nucleotide diversity identified two significant peaks: *rps3-rpl2* (Pi = 0.13533) and *ndhF-rpl32-trnL* (Pi = 0.10533), which are shown in Figure 5B. The first peak, *rps3-rpl2*, reflected the sequence variations that occur in *R. chinensis* (accession No. MG267385). The second one, *ndhF-rpl32-trnL*, indicated the sequence variations occurring in *R. typhina* (accession No. MN866894 and MT083895). These regions can be used as a source of potential barcodes for the identification of the *Rhus* species as well as resources for inferring the phylogenetic relationships of the genus.

The mauve alignment revealed that all six species were relatively conserved. All three species, *R. chinensis*, *R. potaninii*, and *R. typhina*, revealed a syntenic structure, and no large-area and multi-segment gene rearrangement was detected in the cpDNA sequences. In particular, we found that the position of the *ycf15* gene was different in *R. chinensis* than in other *Rhus* species. The *ycf15* gene in the IR region was located between the *rps12* gene and the *trnV-GAC* gene in *R. chinensis*; however, it was positioned between the *ycf2* gene and the *trnL-CAA* gene in the species *R. potaninii* and *R. typhina*, which showed the gene rearrangement.

We compared the exact IR boundary positions and the adjacent genes of six *Rhus* species by IRscope, and the detailed comparison for the four genomic boundaries (LSC/IRA, LSC/IRB, SSC/IRA, and SSC/IRB) are shown in Figure 6. The lengths of the LSC, IR, and SSC regions were similar among the six *Rhus* genomes, and the IR organization was also highly conserved with minor variances for expansions and contractions. The functional *ycf1* gene crossed the IRA/SSC boundary to create the *ycf1* pseudogene fragment at the IRB region in all the genomes, which means that the *ycf1* pseudogene overlapped with the *ndhF* gene in the SSC and IRa junctions with a stretch of 30 to 42 bp. The *trnH* and *rpl2* genes were entirely located within the LSC and IR regions, respectively. The *rps19* gene of *R. chinensis* and *R. potaninii* was entirely located in the LSC region, but it was positioned at the boundary between LSC and IRB in the two *R. typhina* accessions.

There have been many studies and reports on the expansion and contraction of the IR region, and the main view on the mechanism of the slighter IR region expansion is that it may be caused by gene transfer to different regions, while the larger IR region expansion may be realized by a double-strand break repair (DSBR) mechanism [45,46].

Although the chloroplast genome has a nearly collinear gene order in most land plants, the changes in the genome such as gene loss [47], expansion, and contraction at the borders of the IR regions cause size variations [42]. Our results likewise indicate that the downstream sequence of IRb/SSC was conserved and that the *ndhF* gene was adjacent to the IRb/SSC boundary, consistent with the general pattern described in angiosperms [48].

Barrett et al. (2020) found the loss of *rps19* and *rpl22* genes from the chloroplast genome of *R. chinensis* (accession No. MG267385) [49], while the two genes were annotated in our current study. With the acquisition of the new complete chloroplast genome of *R. chinensis*, more complete genomic information indicated that the *rps19* gene was present in the *R. chinensis* sample and that there were some minor differences in its location compared to other *Rhus* species.

As shown in Figure 3C, the SSRs analysis for six *Rhus* species showed that the number of SSRs in *R. typhina* was the highest (92), while in *R. chinensis* it was the lowest (57). These SSRs are similar and divided into five types of microsatellites, i.e., mononucleotide, dinucleotide, trinucleotide, tetranucleotide, and pentanucleotide, most of which consisted of mono- and tri-repeat motifs. However, hexanucleotide repeats were not detected in the six *Rhus* species. The current results are consistent with the recent studies, which exhibited that the SSRs detected in the chloroplast genome of angiosperms were usually composed of A or T repeats and rarely comprised tandem G and C repeats [50]. Therefore, SSRs extended a greater contribution to the ‘AT’ diversity of the *Rhus* chloroplast genomes, as previously reported in different species [51]. These analyses also revealed that approximately 75% of the SSRs were determined in non-coding regions. Microsatellites being distributed throughout the genome, the SSRs in the chloroplast genome can be highly variable at the intra-specific level and may be used as genetic markers in population genetics and evolutionary studies, and in particular, polymorphic SSRs can be used to study genetic diversity, population structure, and biogeography within and between groups [52,53].

### 3.4. Phylogenetic Analysis

The complete chloroplast genomes provided valuable information in plant phylogenies due to their highly conserved structure and higher evolutionary rate when compared to the mitochondrial genome [54]. In recent decades, numerous analyses on the comparison of plastid protein-coding genes [55] and complete genome sequences [56] have been conducted to answer the phylogenetic disposition at deep nodes and the evolutionary relationships among angiosperms were further revealed. In this study, in order to present the relationships among the Anacardiaceae species, we conducted an ML phylogenetic tree based on all the 86 protein-coding genes from the complete chloroplast genomes with *B. sacra* and *C. gileadensis* in the family Burseraceae as outgroups. The multiple alignment of protein-coding sequences possessed 70,234 bp nucleotide sites, of which 4452 were variable and 2796 were parsimony informative. The ML tree is shown in Figure 7. The ML phylogenetic analysis supported the monophyly of each tribe and each genus well, and the samples were largely classified into three groups. The tribe Rhoideae was close to the tribe Anacardieae with a high support of 100%, and they then grouped with the tribe Spondiadeae. In the tribe Rhoideae clade, the genus *Rhus* was sister to *Pistacia* with a high support of 100%, and they then grouped with the *Toxicodendron* species, consistent with the findings from a previous analysis [57]. In the genus *Rhus*, our sample was closely clustered with the three other *R. chinensis* individuals, forming a sister to *Rhus potaninii*, which then grouped with *R. typhina*.

Chloroplast genome sequence has been widely used to reconstruct the phylogenetic relationships among plant lineages [58,59]. The current phylogenetic analysis showed the relationships between the species in the Anacardiaceae family. As an economic and ecological species, the phylogenetic relationship of *R. chinensis* is also a base for further edible and medicinal research. The more complete chloroplast genomes of the *Rhus* species would provide valuable genetic information for the further conservation and evolutionary research.

## 4. Conclusions

In this study, we assembled the complete chloroplast genome sequence of *Rhus chinensis* and identified its basic structure and gene content. After comparing this newly sequenced chloroplast genome with closely related species, we found some differences in genome size, GC content, gene number, and order between species. We also observed the contraction and expansion of the IR boundaries. Our findings demonstrate that most SSRs were A/T rich and located in the intergenic regions (IGS). The analysis of nucleotide divergence of the chloroplast genome sequences exhibited higher levels in the non-coding region than in the coding region. Ten highly variable regions were identified, which will potentially provide plastid markers for further taxonomic, phylogenetic, and population genetic studies in the *Rhus* genus. Furthermore, our analysis also determined gene rearrangement, that is, the position of the *ycf15* gene was different in *R. chinensis* and other *Rhus* species. The phylogenetic trees constructed using all protein-coding genes supported the monophyly of each tribe, and *Rhus* was sister to *Pistacia*, which then grouped with *Toxicodendron*. The results obtained in this study are expected to provide valuable genetic resources to perform species identification, molecular breeding, and intraspecific diversity of the *Rhus* species.

## Figures and Tables

**Figure 1 genes-13-01936-f001:**
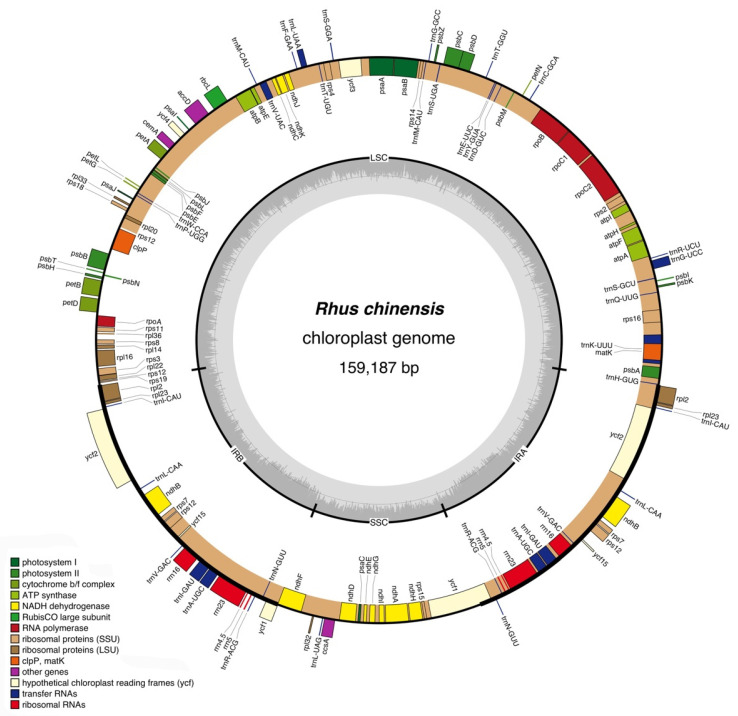
Gene map of the complete chloroplast genome of *R. chinensis*. The genes inside and outside of the circle are transcribed in clockwise and counterclockwise directions, respectively. The light and darker gray in the inner circle correspond to AT and GC content, respectively.

**Figure 2 genes-13-01936-f002:**
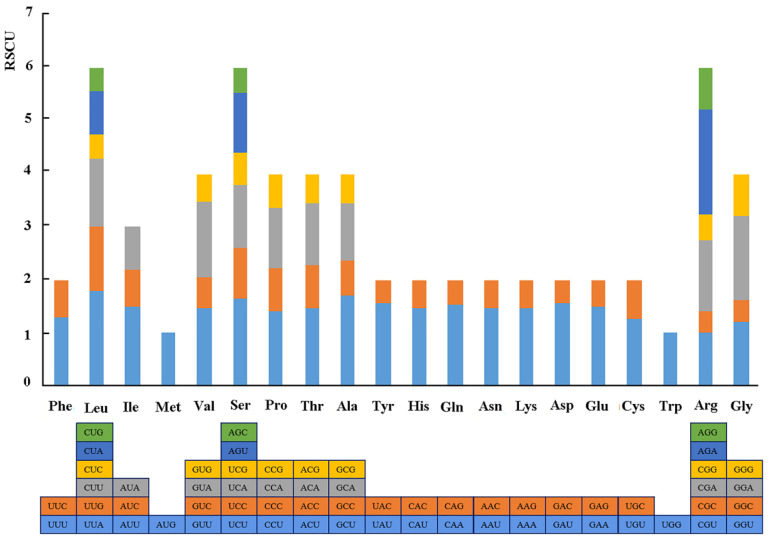
Relative synonymous codon usage (RSCU) of the complete chloroplast genome of *R. chinensis* in this study.

**Figure 3 genes-13-01936-f003:**
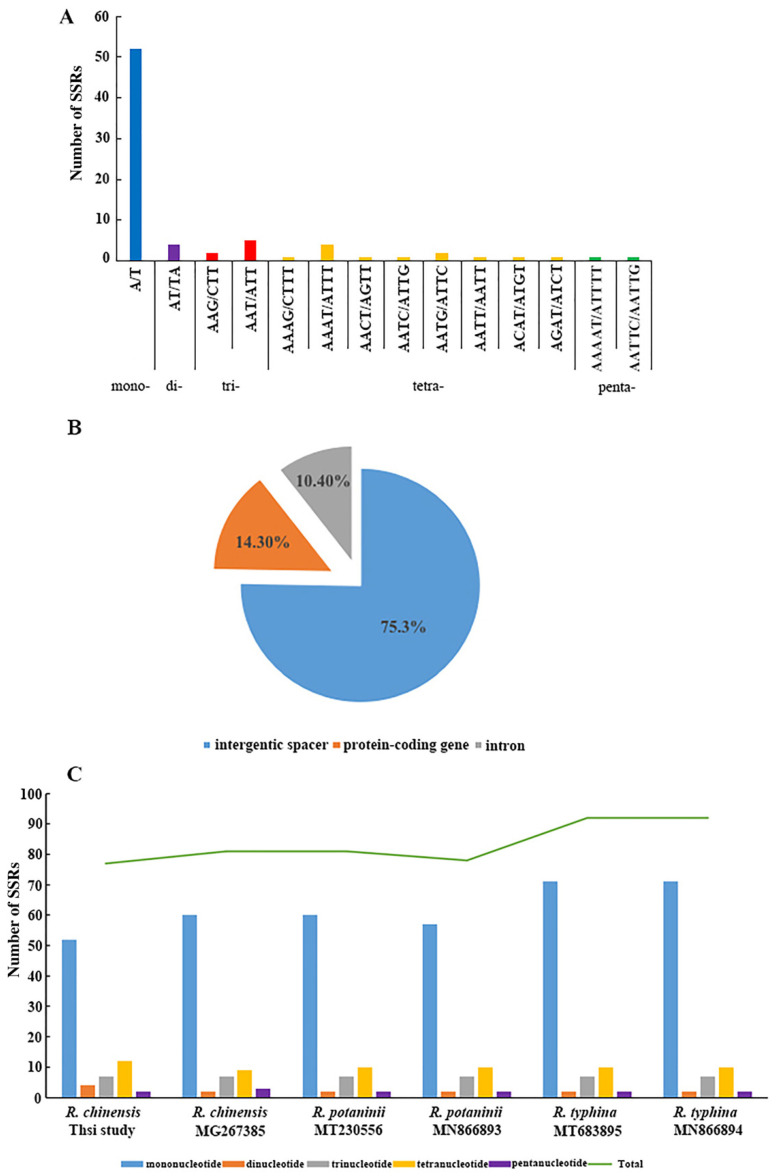
SSRs in the chloroplast genome of *Rhus* species. (**A**) Frequency of identified SSR motifs in *R. chinensis*. Mono-: Mononucleotide, Di-: Dinucleotide, Tri-: Trinucleotide, Tetra-: Tetranucleotide, Penta-: Pentanucleotide, Hexa-: Hexanucleotide. (**B**) Location distribution of all the SSR motifs in *R. chinensis*. (**C**) Number of different SSR types detected in complete chloroplast genomes of *Rhus*.

**Figure 4 genes-13-01936-f004:**
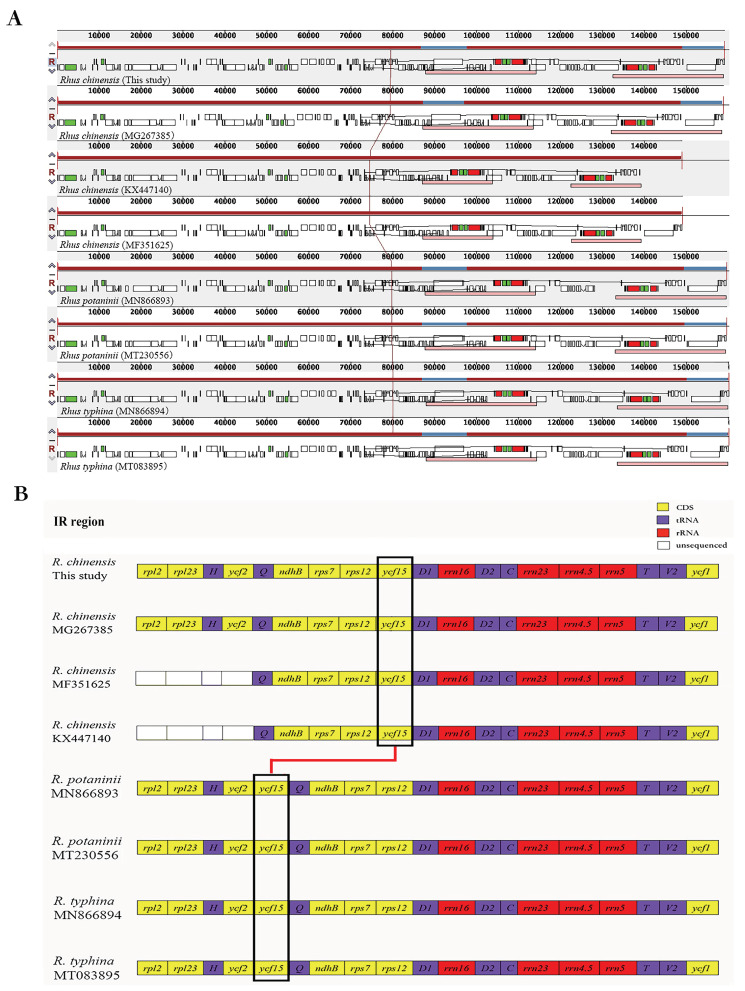
Gene arrangement of eight chloroplast genomes of *Rhus* species. (**A**) Complete genomes. (**B**) IR regions. The gene *ycf15* in the black frame changed position (rearranged) in the chloroplast genome of *Rhus* species.

**Figure 5 genes-13-01936-f005:**
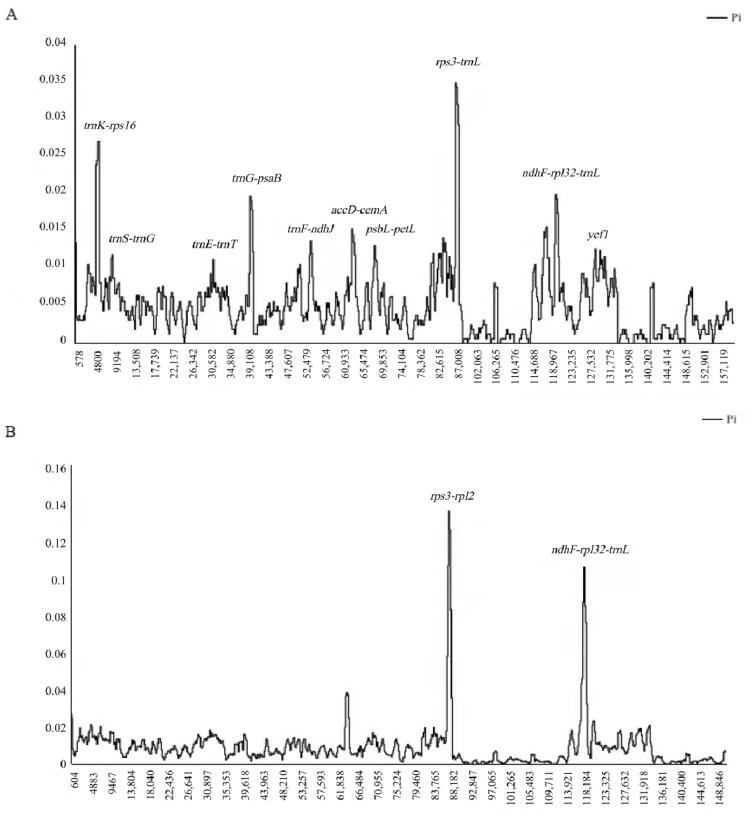
Analysis of nucleotide diversity of the *Rhus* chloroplast genomes. (**A**) Four individuals (window length: 800 bp; step size: 200 bp). X-axis: position of the midpoint of a window; Y-axis: nucleotide diversity of each window. (**B**) Six chloroplast genomes of *Rhus* except *R. chinensis* (accession Nos: KX447140 and MF351625) (window length: 800 bp; step size: 200 bp). X-axis: position of the midpoint of a window; Y-axis: nucleotide diversity of each window.

**Figure 6 genes-13-01936-f006:**
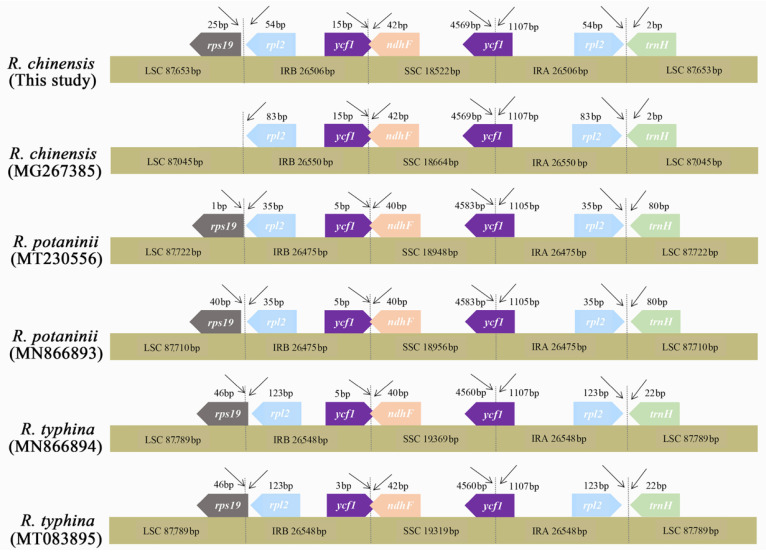
Comparison of the SC/IR junction regions for complete chloroplast genomes of *Rhus* species. The number above the gene features indicates the distance between the ends of the genes and the border sites. Features are not to scale.

**Figure 7 genes-13-01936-f007:**
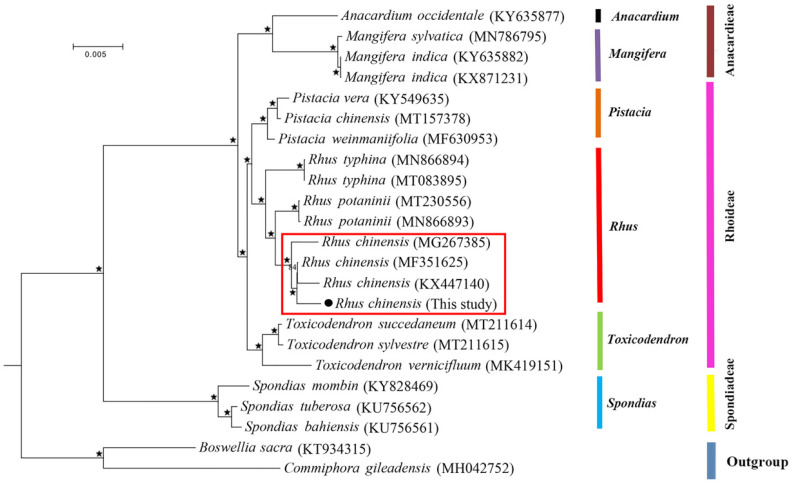
Maximum likelihood (ML) phylogeny of *Rhus* and other species of Anacardiaceae based on all 86 protein-coding genes with *B. sacra* and *C. gileadensis* as outgroups (stars represent nodes with 100% bootstrap values; the clade in the red frame shows the four accessions from the species *R. chinensis*).

**Table 1 genes-13-01936-t001:** Information of complete chloroplast genomes of Anacardiaceae species used in this study.

Species	Family	Accession No.	Size (bp)
*R. chinensis*	Anacardiaceae	This Study	159,187
*R. chinensis*	Anacardiaceae	KX447140	149,011
*R. chinensis*	Anacardiaceae	MF351625	149,094
*R. chinensis*	Anacardiaceae	MG267385	158,809
*R. potaninii*	Anacardiaceae	MT230556	159,620
*R. potaninii*	Anacardiaceae	MN866893	159,616
*R. typhina*	Anacardiaceae	MT083895	160,204
*R. typhina*	Anacardiaceae	MN866894	160,254
*Pistacia chinensis*	Anacardiaceae	MT157378	160,596
*Pistacia vera*	Anacardiaceae	KY549635	160,674
*P. weinmanniifolia*	Anacardiaceae	MF630953	160,767
*Toxicodendron succedaneum*	Anacardiaceae	MT211614	150,710
*Toxicodendron sylvestre*	Anacardiaceae	MT211615	159,600
*Toxicodendron vernicifluum*	Anacardiaceae	MK419151	159,571
*Mangifera indica*	Anacardiaceae	KX871231	157,780
*Mangifera indica*	Anacardiaceae	KY635882	157,780
*Mangifera sylvatica*	Anacardiaceae	MN786795	158,106
*Anacardium occidentale*	Anacardiaceae	KY635877	172,199
*Spondias tuberosa*	Anacardiaceae	KU756562	162,039
*Spondias bahiensis*	Anacardiaceae	KU756561	162,218
*Spondias mombin*	Anacardiaceae	KY828469	162,302
*B. sacra*	Burseraceae	KT934315	160,543
*C. gileadensis*	Burseraceae	MH042752	160,268

**Table 2 genes-13-01936-t002:** Characteristics of the complete chloroplast genomes of *Rhus* species.

Species	*R. chinensis*				*R. potaninii*		*R. typhina*	
Accession No.	OP326720	KX447140	MF351625	MG267385	MN866893	MT230556	MN866894	MT083895
Location	Hubei	Gangwon	Shandong	Anhui	Beijing	Shaanxi	Unknown	Shandong
Size (bp)	159,187	149,011	149,094	158,809	159,616	159,620	160,254	160,204
LSC (bp)	87,653	96,882	97,246	87,045	87,710	87,722	87,789	87,789
SSC (bp)	18,522	18,674	18,644	18,664	18,956	18,948	19,453	19,319
IR (bp)	26,506	16,741	16,602	26,550	26,475	26,475	26,506	26,548
No. of total genes	132	126	126	130	131	133	130	130
Protein-coding genes	86	82	82	85	86	86	85	85
tRNAs	37	36	36	37	37	37	37	37
rRNAs	8	8	8	8	8	8	8	8
Overall GC content (%)	37.8	37.8	37.9	37.9	37.9	37.9	37.8	37.8
GC content of LSC (%)	35.9	36.2	36.2	36.0	36.0	36.0	35.8	35.8
GC content of SSC (%)	32.6	32.7	32.7	32.6	32.6	32.6	32.5	32.5
GC content of IR (%)	42.9	45.4	45.5	42.6	43.0	43.0	43.0	43.0

**Table 3 genes-13-01936-t003:** Lengths of exons and introns of coding genes in the complete chloroplast genomes of four *R. chinensis* accessions.

Gene	Exon Ⅰ (bp)				Intron Ⅰ (bp)				Exon Ⅱ (bp)				Intron Ⅱ (bp)				Exon Ⅲ (bp)			
	1	2	3	4	1	2	3	4	1	2	3	4	1	2	3	4	1	2	3	4
*rps16*	226	226	227	226	894	899	896	882	38	38	40	38								
*atpF*	411				748	748	760	747	156	156	144	156								
*rpoC1*	1626				760	760	760	755	435											
*ycf3*	155				813	797	797	798	226				732	735	733	734	126			
*clpP*	228				631	653	638	637	291	291	292	291	773	776	768	775	69	69	71	69
*petB*	6				785	786	785	772	642											
*petD*	8				741	724	741	750	475											
*rpl16*	402				1068	1071	1068	1133	9											
*rpl2*	393				662	662	662	661	435											
*ndhB*	777				681				756											
*rps12*									232				536				26			
*ndhA*	541				1120	1120	1129	1124	467	467	551	467								
*rps12*	26				536				232											
*ndhB*	756				681				777											
*rpl2*	434	—	—	434	665	—	—	664	391	—	—	391								
*trnK-UUU*	35				2599	2593	2596	2596	37											
*trnG-UCC*	23				714	713	713	713	47											
*trnL-UAA*	37				450				50											
*trnV-UAC*	37				585				39											
*trnA-UGC*	35				841				38											
*trnI-GAU*	35				950	949	950	940	42											
*trnI-GAU*	42				950	949	950	940	35											
*trnA-UGC*	38				841				35											

Notes: 1, This study; 2, KX447140; 3, MF351625; 4, MG267385.

**Table 4 genes-13-01936-t004:** Variable site of complete chloroplast genomes of four *R. chinensis* accessions and six *Rhus* species.

Species	Region	Total Sites	Variable Sites	Informative Sites	Nucleotide Diversity
*R. chinensis*	Large single-copy region	98,636	969	29	0.00584
	Small single-copy region	18,751	281	20	0.00783
	Inverted repeat region	27,478	48	11	0.00156
	Complete cp genome	160,884	1436	93	0.00502
Six *Rhus* species	Large single-copy region	89,280	1743	1075	0.00958
	Small single-copy region	19,572	616	435	0.01614
	Inverted repeat region	27,432	109	61	0.00184
	Complete cp genome	162,196	2931	1621	0.00841

## Data Availability

The data which supports the findings of this study are openly available in GenBank of NCBI at https://www.ncbi.nlm.nih.gov (accessed on 30 August 2022), GenBank accession number: No. OP326720.

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
