# Peer review of "Variation among the Complete Chloroplast Genomes of the Sumac Species *Rhus chinensis*: Reannotation and Comparative Analysis"

_genes, 2022, doi:10.3390/genes13111936_

Round 1

Reviewer 1 Report

The paper Variation Among the Complete Chloroplast Genomes of the Sumac Species Rhus chinensis: Re-annotation and Comparative Analysis by Yujie Xu , Xu Su and Zhumei Ren gives a complete and valuable information about the chloroplast genome of Rhus chinensis, an species with some agricultural, medical and ecological interest. The main flaw that can be found in this article is the comparison of their actual results with previous annotations of the complete chloroplast genome of the same species. The variation among them is too wide to be credible. There are probably gross errors in some of the annotations.  This might be discussed in the article and should be recommendable shorten the matter devoid to these intraspecific comparisons.As the authors write: "However, the variation of the three genomes is so great no matter for the length or the gene organization" (a sentence that should be rewrtten in a clearer form).

Minor points:

Line 52: "Data set" is repeated.

Table 2 (pages 4-5): the table is broken and divided in two pages. Under MN866894, must be written "Unknown", not "Unknow".

Lines 175-76: mat K is a gene, which shows some divergence among chloroplast genomes. Therefore, the sentence "matK and was the largest intron with a length of 2599 bp, which has similar characteristics with other land plants " must be rewritten in a more precise and informative style.

Lines 185-87: I do not understand the following sentence: "...all ended with A/U except UUG AUG and UGG, of which about 42.86% ended with A and about 57.14% ended with U, indicating these codons tend to end with A/U."

Figure 3B: The word "intro" must be completed as "intron"

Line 219: Korea, not "Kerea"

Lines 244-246. The sentence must be rewritten.

Figure 4. The legend of this figure is not complete. What does mean the boxes?

Reviewer 2 Report

This manuscript presents the outcome of a research that includes the full plastome sequence of Rhus chinensis. Then, the platome is described thoroughly and a phylogenetic comparison is performed against other species of the same genus and the same family.

I have some problems with the scope of the study. Three plastomes of the same species have been sequenced in the past and these sequences are available in the databases. What is the reason for sequencing one more individual? What is the novelty of this study and the added value from this research? This is not clear in the text. Another thing that is not clear is the phylogenetic question of the comparison of the plastome sequences among the specific taxa. Which hypothesis is tested? Why were these taxa chosen? Is there any evolutionary conclusion from this comparison?

Also, the presentation of this manuscript is not of sufficient quality: English should be corrected, especially syntax. Words are double in the sentences, some others are missing. Some words start with capital letters for no reason. A native speaker or a language specialist should check this manuscript before further consideration (if any).

Here are some specific remarks.
Line 21: This is a very unusual way to present this kind of information, especially in the abstract. I would prefer to have it explained in words.
Line 32: "that widely distributed" maybe "that is widely distributed"
Lines 34-35: properties are not medicinal purposes.
Line 37: Instead of Rhus chinensis, better R. chinensis for the rest of the manuscript. And why is the name of the species two times in the sentence?
Lines 68-71: I do not understand the meaning here. What is wrong with the sequenced genomes?
Lines 95-97Q I understand that there were two methods used for the assembly: de novo with GetOrganelle and using a reference plastome KX447140 of the same species. Which of the two genomes obtained was used in the next steps of the analysis?
Line 98: DOGMA is abandoned by its developers and is obsolete. When did this analysis occur?
Line 137: Why were only protein-coding genes used for phylogenetic studies?
Line 222: existed?
Line 391: "...the mysterious evolutionary relatedness..." is not a nice expression!
Lines 399-400: This type of presentation is unusual. I believe that this information is in the phylogenetic tree that follows and this can be described with words.
Line 407: "has been wildly used"....

Reviewer 3 Report

The article “Variation Among the Complete Chloroplast Genomes of the Sumac Species Rhus chinensis: Re-annotation and Comparative Analysis”, presents the sequencing and assembly of the plastid genome of R. chinensis and the comparisons with other R. chinensis and Rhus pt genomes. The article shows a detailed analysis of several characteristics of pt genomes as GC content, number of genes, number of introns, between others and the comparison with other samples of the same species. The most interesting result is the detection of the hotspot of variability, as pt genome is useful as a marker for phylogeny analysis”.

Some comments for authors:

Introduction:

It is not clear why was relevant to obtain the complete sequences of a chloroplast genome from a species that already have its chloroplast genome sequenced.

What is the meaning of compiled? Line 72.

Results:

Figure 4a must to be improved. The quality of the figure is poor. Why are there lines connecting right blocks while left blocks remain unconnected?

The results section is extensive. Sometimes repetitive, line 180 should be only in materials and methods.

Line 185, there is a strange symbol followed by UUG.

The paragraph relative to RSCU could be shortened. There is too much information that is not relevant.

In Figure 2, the axis label RSCU is in the wrong direction.

Figure 3 A. the y-axis needs a label. Figure 3 b. What is the meaning of “intro”? Figure 3 C. the y-axis label is incorrect.

Section 3.2 from lines 217 to 221 is redundant.

Lines 233 to 236 are redundant with lines 242 to 244.

Line3s 246 to 248 are redundant with lines 234 to 235.

The legend of Figure 5 does not match with the analyses “. Sliding window analysis of the complete chloroplast genome of Rhus chinensis”. It must be “Analysis of nucleotide diversity”

Line 356 “angiospermia”

Some points need to be clarified.

The region not present in the assembly must to be corroborated to present a more reliable sequence.

Some figures could be supplementary.

Round 2

Reviewer 1 Report

The paper has been sufficiently improved to be accepted after a minor revision of phrasing. For instance, the sentence: "The galls are rich in tannins and a source of traditional Chinese medicine and economically very important for the extracted tannins extracted from the galls have been utilized earlier", must be rewritten. But this revision must be extended to the full paper.

The line numbers cited by the authors in their answers are not the actual line numbers in the revised ms. This makes difficult to follow changes performed in the original paper.
